# MORPH-LER: Log-Euclidean Regularization for Population-Aware Image Registration

**Mokshagna Sai Teja Karanam**[*1,2]                    MOKSHAGNA.KARANAM@UTAH.EDU

**Krithika Iyer**[*1,2]                                 KRITHIKA.IYER@UTAH.EDU

**Sarang Joshi**[2,3]                                   SARANG.JOSHI@UTAH.EDU

**Shireen Elhabian**[1,2]                               SHIREEN@SCI.UTAH.EDU

[1] *Karlert School of Computing, University of Utah, UT, USA*

[2] *Scientific Computing and Imaging Institute, University of Utah, UT, USA*

[3] *Biomedical Engineering Department, University of Utah, UT, USA*

**Editors:** Accepted for publication at MIDL 2025

## Abstract

Spatial transformations that capture population-level morphological statistics are critical for medical image analysis. Commonly used smoothness regularizers for image registration fail to integrate population statistics, leading to anatomically inconsistent transformations. Inverse consistency regularizers promote geometric consistency but lack population morphometrics integration. Regularizers that constrain deformation to low-dimensional manifold methods address this. However, they prioritize reconstruction over interpretability and neglect diffeomorphic properties, such as group composition and inverse consistency. We introduce MORPH-LER, a Log-Euclidean regularization framework for population-aware unsupervised image registration. MORPH-LER learns population morphometrics from spatial transformations to guide and regularize registration networks, ensuring anatomically plausible deformations. It features a bottleneck autoencoder that computes the principal logarithm of deformation fields via iterative square-root predictions. It creates a linearized latent space that respects diffeomorphic properties and enforces inverse consistency. By integrating a registration network with a diffeomorphic autoencoder, MORPH-LER produces smooth, meaningful deformation fields. The framework offers two main contributions: (1) a data-driven regularization strategy that incorporates population-level anatomical statistics to enhance transformation validity and (2) a linearized latent space that enables compact and interpretable deformation fields for efficient population morphometrics analysis. We validate MORPH-LER across two families of deep learning-based registration networks, demonstrating its ability to produce anatomically accurate, computationally efficient, and statistically meaningful transformations on the OASIS-1 brain imaging dataset. https://github.com/iyerkrithika21/MORPH_LER

**Keywords:** deformable image registration, manifold statistics, non- rigid, diffeomorphisms, shape population statistics, log- euclidean statistics

## 1. Introduction

Image registration is crucial in medical image analysis. It entails determining a one-to-one mapping of pixel coordinates between images to align corresponding anatomical points. Registration algorithms establish correspondences that enable the creation of population-level atlases, providing standardized references for studying disease progression, pathology detection, treatment planning, and motion tracking (Suganyadevi et al., 2022; Zachiu et al.,

---

* Contributed equally

2020; Binte Alam et al., 2020; Viergever et al., 2016). The versatility of image registration methods makes them indispensable across various imaging modalities (Huang et al., 2022). Diffeomorphisms are smooth, invertible spatial transformations that preserve image topology and are a key class of spatial transformations. They prevent artifacts like tearing or folding, enabling precise comparisons of anatomical structures across patients and time points. Their flexibility makes them crucial in medical imaging, particularly for capturing complex deformations, aiding early diagnosis and treatment. Traditional methods like Large Deformation Diffeomorphic Metric Mapping (LDDMM) (Beg et al., 2005; Joshi and Miller, 2000), optical flow (Zhai et al., 2021), Direct Deformation Estimation (DDE) (Boyle et al., 2019) optimize for transformations using metric distances or pixel movement vectors. Despite their effectiveness, these methods struggle with large deformations, inter-subject anatomical variability, and computational efficiency. Pairwise registration often biases results toward the reference image and struggles to deal with significant anatomical differences.

Deep learning methods address these challenges by learning complex transformations directly from images, offering faster inference and ensuring smooth deformation fields through regularizers like bending energy or $L_2-$gradient penalties (Balakrishnan et al., 2019; Chen et al., 2022). However, they treat registration as a computational problem, neglecting diffeomorphic properties like inversion and composition, resulting in biologically implausible solutions. Improvements like gradient and transformation inverse consistency enhance diffeomorphic transformations (Tian et al., 2023). Yet, existing approaches often predict transformations without deeper insights into underlying anatomical relationships, oversimplifying interpretations.

Nevertheless, the mathematical framework of diffeomorphic transformations (smooth, invertible transformations that preserve the topology of the image, meaning they don't tear or fold it) defined by the diffeomorphism group $Diff(\mathcal{M})$ (the set of all such transformations), which, mathematically, has the structure of a Fréchet Lie group and manifold provides a rigorous foundation for capturing and analyzing complex anatomical variability. Understanding this framework is essential for ensuring image registration methods produce realistic and meaningful results.

Unlike Euclidean spaces, operations like addition or averaging cannot be directly defined on the diffeomorphism group $Diff(\mathcal{M})$. This means we can't simply average two diffeomorphic transformations to get a mean transformation in the same way we can average two points in a Euclidean space. Techniques such as Principal Geodesic Analysis (PGA) (Fletcher et al., 2004), Fréchet means (Le and Kume, 2000), and geodesic regression (Fletcher, 2011) attempt to address statistical estimation for Riemannian manifolds (curved spaces where distances and angles can be measured), but require significant adaptation for Fréchet Lie groups. These techniques need to be modified to account for the complex structure of diffeomorphism groups. Therefore, alternative metrics are necessary to model relationships between deformation fields (the spatial displacements that map one image to another) and ensure anatomical relevance. The Log-Euclidean framework (Arsigny et al., 2006) transforms non-linear diffeomorphism manifolds into linear Lie algebra spaces (vector spaces that approximate the diffeomorphism group locally), enabling efficient computation of transformation distances and statistical analysis. However, iterative processes used in logarithm estimation can be susceptible to initializations, potentially leading to subopti-

mal results. Standalone deformation analysis models address the limitations of the iterative methods and provide a computationally efficient transformation of diffeomorphic groups into linear Lie algebra. By mapping the problem to a Lie algebra, we can perform calculations more easily. These mathematical foundations provide a rigorous framework for analyzing the complex structure of diffeomorphisms, leading to more reliable and interpretable results.

However, a unified approach is needed to bridge computational methods for generating diffeomorphisms and statistical techniques for their analysis. To address this gap, we introduce MORPH-LER, a Log-Euclidean regularization framework for population-aware unsupervised image registration. By leveraging population morphometrics, MORPH-LER guides and regularizes registration networks while enforcing inverse consistency in latent and diffeomorphic spaces, enabling data-driven regularization for valid transformations and enhanced morphological analysis. We validate MORPH-LER as a plug-and-play regularizer for deep learning registration networks and demonstrate the model's efficiency, robustness, and broad applicability.

## 2. Related Work

Traditional image registration methods align images by optimizing similarity metrics like mutual information, normalized cross-correlation, or the sum of squared differences. While accurate, they are time-consuming, particularly for high-resolution and large-scale datasets. Diffeomorphic (smooth and invertible) transformations are popular as they preserve anatomical topology in deformation analysis (Fu et al., 2020). However, these approaches are computationally expensive, limiting scalability due to the need to enforce smoothness, invertibility, and anatomical consistency. Deep learning methods have emerged as faster and scalable alternatives. Early models used convolutional networks like U-Net (Ronneberger et al., 2015) to predict deformation fields directly, reducing computational time (Boveiri et al., 2020). However, their limited receptive fields led to the adoption of transformer-based models, which better capture global spatial relationships (Chen et al., 2022). Despite these advancements, both approaches rely on smoothness regularization, leading to anatomically inconsistent deformations.

Independent efforts have advanced deformation field analysis tools. PGA (Fletcher et al., 2004; Fletcher, 2011) extends Principal Component Analysis (PCA) to Riemannian manifolds. While PGA offers a rigorous approach to analyzing deformations on Riemannian manifolds, defining an appropriate Riemannian metric for complex anatomical variations remains challenging. The Log-Euclidean framework (Arsigny et al., 2006) provides a computationally efficient approach to perform statistics on diffeomorphisms by mapping them to a linear vector space via principal logarithms. This framework enables using Euclidean operations on logarithms, preserves key mathematical properties, and simplifies computations compared to Riemannian approaches. However, estimating principal logarithms using the iterative method is computationally expensive, particularly for high-dimensional data or large deformations.

Population-aware registration networks, such as CAE (Bhalodia et al., 2019), add regularization by limiting deformation fields to low-dimensional manifolds, effectively capturing anatomical variability. However, they often fall short of explicitly enforcing diffeomorphism, essential for interpretability and robustness. CAE's regularization approach begins with an L2 regularizer, providing initial stability. However, when the autoencoder-based

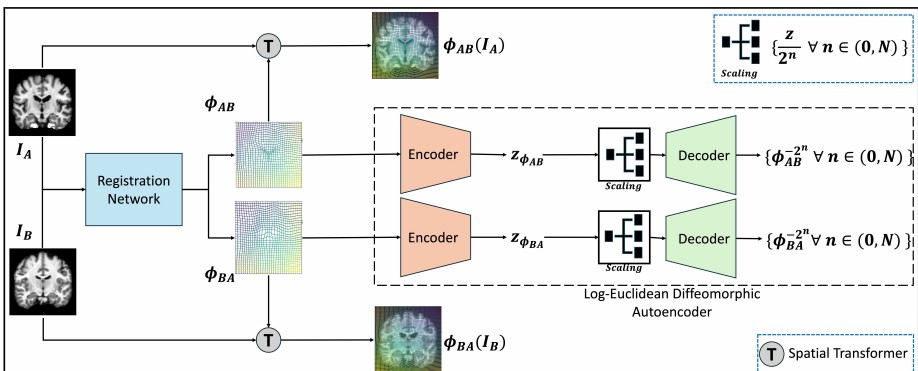

Figure 1: Proposed Architecture for MORPH-LER

regularizer is introduced, it struggles to preserve anatomical consistency, often resulting in non-diffeomorphic transformations.

Similarly, diffeomorphic autoencoders for LDDMM (Hinkle et al., 2018) have been proposed for atlas building by encoding velocity field-based transformations into latent spaces for statistical analysis. These methods face computational challenges with large datasets and complex anatomical variations. Additionally, the focus on deformation field reconstruction from latent space representation often sacrifices interpretability, limiting their utility for downstream tasks such as population analysis. The Log Euclidean Diffeomorphic Autoencoder (LEDA) (Iyer et al., 2024) framework has combined the advantages of the Log Euclidean framework with a deep learning approach. LEDA estimates the principal logarithms of the deformation field while maintaining inverse consistency in both the deformation and latent spaces. This approach efficiently captures complex anatomical variations while remaining computationally tractable for large-scale analysis. It maps composition in the data space to linearized latent space, enabling vector-based statistics and ensuring robustness while adhering to diffeomorphism group laws.

To address the crucial gap in integrating population-level statistics with image registration models in an end-to-end manner, we propose MORPH-LER, a Log-Euclidean population-driven image registration framework. By leveraging collective anatomical insights, MORPH-LER ensures anatomically meaningful transformations while maintaining computational efficiency.

## 3. Methods

**Background:** A diffeomorphism $\phi : \mathbb{R}^D \to \mathbb{R}^D$, is a smooth, invertible mapping with a smooth inverse, defined as $\phi(\mathbf{x}) = \mathbf{x} + \mathbf{u}(\mathbf{x})$, where $\mathbf{u}(\mathbf{x})$ is the displacement field. Diffeomorphic transformations can be constructed by composing multiple small deformations: $\phi(\mathbf{x}) = (\mathbf{x} + \epsilon\mathbf{v}_1(\mathbf{x})) \circ (\mathbf{x} + \epsilon\mathbf{v}_2(\mathbf{x})) \circ \cdots \circ (\mathbf{x} + \epsilon\mathbf{v}_n(\mathbf{x}))$. The space of diffeomorphisms $\mathrm{Diff}(\mathcal{M})$ forms a Lie group under composition, with the identity map serving as the group's identity element; i.e., $\phi \circ \mathbf{id} = \mathbf{id} \circ \phi = \phi, \forall \phi \in \mathrm{Diff}(\mathcal{M})$. The associated Lie algebra consists of smooth vector fields $\mathbf{v}(\mathbf{x}) : \mathbb{R}^D \to \mathbb{R}^D$, which generate diffeomorphisms through their flows, described by one-parameter subgroups $\{\phi_t\}_{t \in \mathbb{R}}$. The exponential map $\exp(\mathbf{v})$ connects the Lie algebra to the Lie group, while the logarithm map $\log(\phi)$ serves as its local inverse. To compute the logarithm map, a non-linear inverse scaling and square rooting algorithm

([Arsigny et al., 2006](#)) is used:

$$\log(\boldsymbol{\phi}) = 2^N \log(\boldsymbol{\phi}^{-2^N}) \quad \text{where} \quad \log(\boldsymbol{\phi}^{-2^N}) = \boldsymbol{\phi}^{-2^N} - \mathbf{id} \tag{1}$$

**MORPH-LER:** The proposed model MORPH-LER is illustrated in Figure [1](#) combines a primary network performing registration task and the secondary network functioning as a regularizer constrains the solution space of the primary task to ensure anatomically meaningful transformations.

*Primary Registration Network* can be any registration module that is designed to produce deformation fields. Given a pair of images A and B, the primary network is tasked with learning two displacement fields: $\boldsymbol{\phi}_{AB}, \boldsymbol{\phi}_{BA}$ where $\boldsymbol{\phi}_{AB}$ corresponds to the warp that ideally should match image A to image B, while $\boldsymbol{\phi}_{BA}$ represents the inverse transformation from B to A. The network produces these displacement fields simultaneously to enable inverse consistency regularization. The displacement fields and their respective source images are passed through a spatial transform unit to produce registered images. The primary network incorporates inverse consistency regularization to ensure reliable bi-directional mappings: $\boldsymbol{\phi}_{AB} \cdot \boldsymbol{\phi}_{BA} = id, \boldsymbol{\phi}_{BA} \cdot \boldsymbol{\phi}_{AB} = id$. This constraint encourages the network to learn transformations that are as close to being inverses of each other as possible, improving the overall accuracy of the registration process. The loss function for the primary network includes: (a) similarity loss: $L_{sim} = SIM(A, \boldsymbol{\phi}_{AB} \circ B) + SIM(B, \boldsymbol{\phi}_{BA} \circ A)$ and (b) inverse consistency loss: $L_{reg} = \|\boldsymbol{\phi}_{AB} \cdot \boldsymbol{\phi}_{BA} - id\|^2 + \|\boldsymbol{\phi}_{BA} \cdot \boldsymbol{\phi}_{AB} - id\|^2$. The total loss is a weighted sum of these components where $\lambda$ represents the weight for the term:

$$L_P = \lambda_{sim} L_{sim} + \lambda_{reg} L_{reg} \tag{2}$$

*Secondary Population-based Regularization Network* uses Log Euclidean Diffeomorphic Autoencoder ([Iyer et al., 2024](#)), which is strongly rooted in the Log-Euclidean statistics framework as the population-based regularizer. LEDA predicts $N$ successive square roots of the deformation field, enabling accurate and computationally efficient logarithmic approximations using equation [1](#). The encoder-decoder architecture is defined as: $f_\gamma(\phi) = z$, $\quad g_\theta\left(\frac{z}{m}\right) = \phi^{1/m}$, where $m = 2^n$, $n \in \{0, 1, \dots, N\}$. LEDA uses the following loss terms:

*Reconstruction Loss:* The deformation field must be accurately reconstructed from its predicted roots. If $\boldsymbol{\phi}^{-m}$ is the predicted root at stage $n$ where $m = 2^n$, the deformation field, when composed $m$ times, must match the original deformation field, the reconstruction loss is $\mathcal{L}_{rec} =$

$$\sum_k \sum_n \left\| C_m(\widehat{\boldsymbol{\phi}}_{AB}^{-m}) - \boldsymbol{\phi}_{AB} \right\|^2 + \left\| C_m(\widehat{\boldsymbol{\phi}}_{BA}^{-m}) - \boldsymbol{\phi}_{BA} \right\|^2 \quad \text{where} \quad C_m(\widehat{\boldsymbol{\phi}}^{-m}) = \underbrace{\boldsymbol{\phi} \circ \cdots \circ \boldsymbol{\phi}}_{m \text{ times}} \approx \boldsymbol{\phi}. \tag{3}$$

*Inverse Consistency Losses:* Even though the primary network enforces inverse consistency, we also want the successive estimated roots of the forward and inverse deformation fields to compose to identity. This is enforced through the inverse consistency loss at each root approximation stage. Along with the deformation field inverse consistency, we also want the latent representations of forward and inverse transformations to be consistent, with equal magnitude but opposite directions: $\mathbf{z}_{AB} = -\mathbf{z}_{BA}$. This is enforced through a latent

Table 1: Registration Performance Metrics: **Top-performing model** and next-best model.

| | | | | Class-Wise Dice Score ↑ | | |
| Methods ↓ | $|\mathbf{J}| < 0\% \downarrow$ | Dice Score ↑ | Cortex | Subcortical-Gray-Matter | White-Matter | CSF |
|---|---|---|---|---|---|---|
| CAE | $25.810 \pm 0.901$ | $0.630 \pm 0.088$ | $0.559 \pm 0.029$ | $0.582 \pm 0.023$ | $0.742 \pm 0.016$ | $0.638 \pm 0.172$ |
| TransMorph | $2.612 \pm 0.494$ | $\mathbf{0.676 \pm 0.054}$ | $\mathbf{0.649 \pm 0.029}$ | $\mathbf{0.590 \pm 0.022}$ | $\mathbf{0.800 \pm 0.019}$ | $\mathbf{0.664 \pm 0.146}$ |
| M-TransMorph | $\mathbf{0.495 \pm 0.174}$ | $0.630 \pm 0.006$ | $0.559 \pm 0.029$ | $0.582 \pm 0.023$ | $0.742 \pm 0.016$ | $0.638 \pm 0.172$ |
| GradICON | $\mathbf{0.058 \pm 0.077}$ | $0.588 \pm 0.049$ | $0.480 \pm 0.025$ | $0.557 \pm 0.026$ | $0.693 \pm 0.017$ | $0.622 \pm 0.129$ |
| M-GradICON | $0.313 \pm 0.185$ | $0.631 \pm 0.052$ | $0.539 \pm 0.027$ | $0.587 \pm 0.023$ | $0.741 \pm 0.015$ | $0.655 \pm 0.145$ |

inverse consistency loss that combines cosine similarity $\Theta_k$ between $\mathbf{z}_{AB}, \mathbf{z}_{BA}$ and magnitude constraints.

$$\mathcal{L}_{inv} = \sum_k \sum_n \left\| \widehat{\phi}_{AB}^{-m} \circ \widehat{\phi}_{BA}^{-m} - \mathbf{id} \right\|^2 \text{ and } \mathcal{L}_{linv} = \sum_k \left( \frac{1 + \cos(\Theta_k)}{2} + \|\mathbf{z}_{AB} + \mathbf{z}_{BA}\|^2 \right) \quad (4)$$

The total loss function for LEDA is given by :

$$\mathcal{L}_S = \alpha_{rec}\mathcal{L}_{rec} + \alpha_{inv}\mathcal{L}_{inv} + \alpha_{linv}\mathcal{L}_{linv}. \quad (5)$$

The total loss function for training the proposed model is $\mathcal{L}_{total} = \mathcal{L}_P + \lambda_1 \mathcal{L}_S$. The training process for MORPH-LER follows a two-phase strategy. Initially, we set $\lambda_1 = 0$ (no LEDA regularizer), allowing the primary network to learn forward and backward diffeomorphic mappings. Subsequently, we activate the LEDA regularizer $\lambda_1 > 1$, enabling simultaneous training of both primary and secondary networks. This approach ensures the primary network learns diffeomorphic transformations before introducing population-based regulation, leading to more accurate and anatomically consistent results.

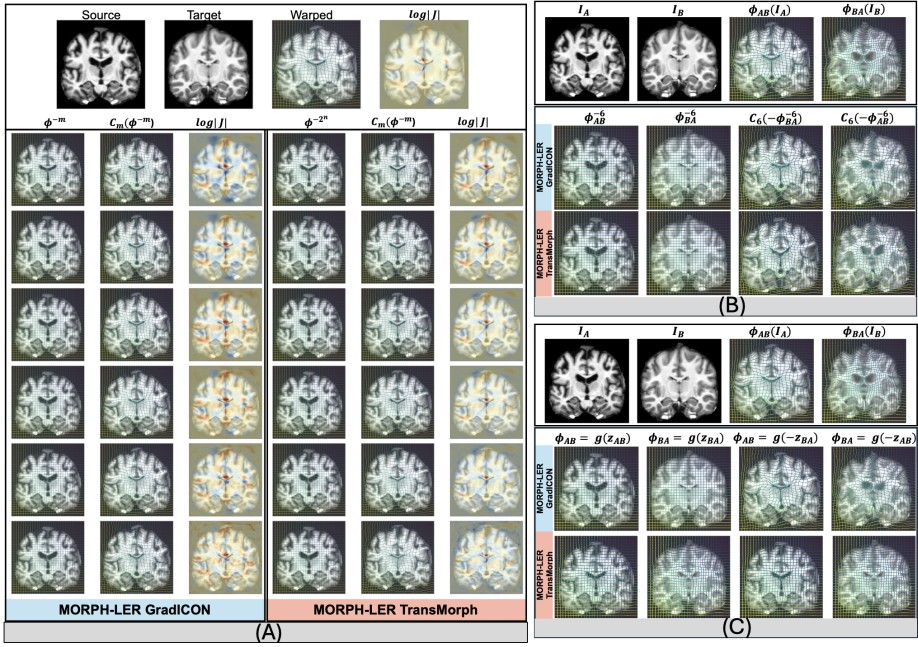

Figure 2: (A) MORPH-LER Square Root Estimation: GradICON (left) and TranMorph (right) as the primary registration network. (B) Validation of Small Deformation Field Assumption. (C) Validation of Latent Inverse Consistency.

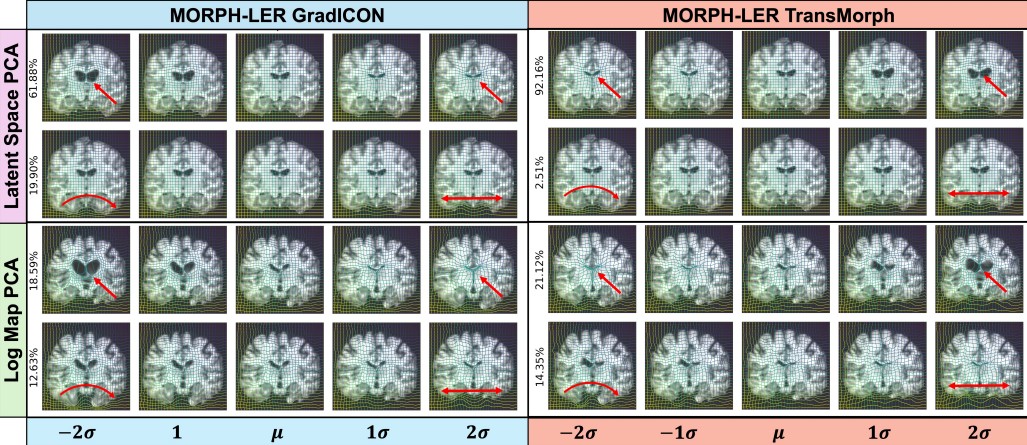

Figure 3: MORPH-LER PCA Modes of Logarithm Maps and Latent Representations. Red arrows highlight key structural shape changes for each PCA mode. We display the deformation field and apply it to a randomly chosen image to illustrate modes of variation in the population of the deformation field.

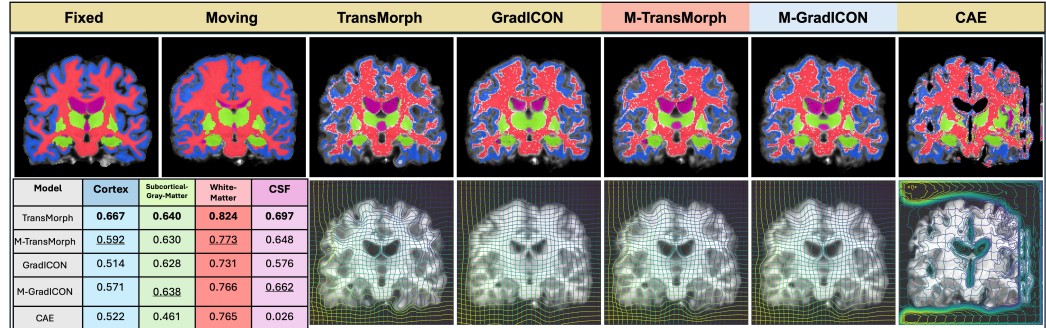

Figure 4: Qualitative Registration Results: Four-label brain segmentation (cortex, gray matter, sub-cortical gray matter, CSF) used to assess registration quality. The table shows Dice scores between warped and target segmentations for the visualized sample, with color-coding matching segmentation regions. Top (bold) and runner-up (underlined) models are highlighted for the visualized sample.

## 4. Results

We use 2D coronal slices from the OASIS-1 dataset (Marcus et al., 2007), which includes brain MRIs from 100 subjects (60 with Alzheimer's). We evaluate our proposed regularizer against several baselines: CAE (Bhalodia et al., 2019), an autoencoder-based method, and the original GradICON (Tian et al., 2023) and TransMorph (Chen et al., 2022) without LEDA. GradICON penalizes deviations from the Jacobian of the inverse consistency constraint, while TransMorph combines a Swin-Transformer (Liu et al., 2021) encoder and a convolutional decoder for volumetric medical image registration. We utilize GradICON and TransMorph as primary registration networks with LEDA as the secondary network, naming them M-GradICON and M-TransMorph. Table 1 compares TransMorph vs. M-TransMorph, GradICON vs. M-GradICON, and CAE as a standalone regularizer. Within each subgroup, we analyze trade-offs between registration accuracy and topological preservation. M-TransMorph improves diffeomorphic properties by reducing negative Jacobian

pixels, but slightly impacts segmentation performance. M-GradICON achieves higher Dice scores than GradICON while maintaining low negative Jacobian pixels, balancing accuracy and topology preservation. Compared to CAE, LEDA-based regularization maintains similar Dice scores while significantly reducing negative Jacobian pixels, demonstrating its superiority in preserving anatomical topology without compromising accuracy. The improvements in metrics achieved by MORPH-LER variants were statistically significant, determined by paired t-tests ($p < 0.05$) across all model comparisons.

Figure 4 showcases a simplified 4-label brain segmentation used to assess registration quality. The segmentation comprises cortex, gray matter, subcortical gray matter, and cerebrospinal fluid, providing a concise yet informative representation of key brain structures. We apply learned deformation fields to these segmentation labels to evaluate registration performance and compute Dice scores between the warped and target segmentations. These results corroborate the quantitative findings. Notably, our proposed M-GradICON model significantly improves over its baseline counterpart. This underscores the effectiveness of the secondary network. The performance of M-GradICON is especially noticeable in the smaller anatomical regions, such as subcortical gray matter and cerebrospinal fluid. The quantitative and qualitative results show that the CAE regularization strategy with a simple autoencoder is ineffective for complex shapes, lacks diffeomorphic properties, and produces anatomically inconsistent transformations. Appendix Figure 6 shows more examples of CAE on toy dataset.

Figure 2.A illustrates the square root estimations $\phi^{-m}$ of MORPH-LER variants and demonstrates a progressive warping of the source image to align with the target. M-GradICON exhibits superior performance over M-TransMorph, featuring smoother deformation grids, better alignment of warped images, and smoother Jacobian maps, indicating diffeomorphic superiority. To validate the small deformation field assumption[1], we tested the logarithm map consistency by utilizing the $2^6$-th root estimation, we systematically negated and composed forward and inverse displacement fields. Figure 2.C demonstrates that both methods satisfy this fundamental assumption, validating the accuracy of logarithm maps.

Furthermore, we extended this validation to the model's latent space by negating the latent representations of forward and inverse displacement fields. As illustrated in Figure 2.B, the MORPH-LER accurately decodes these negated latent representations into their corresponding inverse fields, providing compelling evidence of inverse consistency in the latent space. PCA analysis of the MORPH-LER's latent space, depicted in Figure 3, reveals modes of variation that align closely with logarithm map PCA results. Red arrows highlight key structural shape changes for each PCA mode. These modes capture clinically consistent changes, such as ventricular expansion and hippocampal atrophy, while maintaining smooth, structured transitions. This latent space representation not only supports accurate reconstruction of deformations but also enables intuitive exploration of clinically meaningful variations, highlighting the MORPH-LER's potential for generating realistic deformations, interpolating between anatomical states, and providing valuable insights for understanding disease progression.

---

1. Small deformation field assumption $u_{AB}(\mathbf{x}) = u_{BA}(\mathbf{x})$ reflects symmetry of infinitesimal displacements, ensuring consistent and reversible transformations.

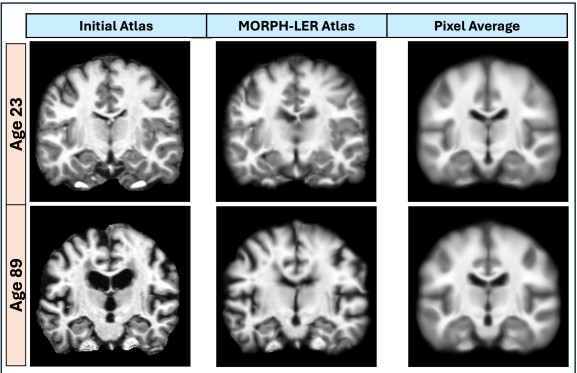

Figure 5: Estimated atlas using the proposed algorithm compared with pixel-wise average atlas.

To create a population-level representation that describes the average shape, and structure, of a population of anatomical objects (Joshi et al., 2004) we propose an efficient atlas estimation approach that leverages the trained MORPH-LER's linearized latent space, which adheres to Lie group action laws. The algorithm begins by randomly selecting an initial image $\mathbf{A}^{(0)}$ as the starting atlas. For each image $\mathbf{I}_i$ in the dataset, the algorithm computes bidirectional transformations ($\boldsymbol{\phi}_{\mathbf{A}^{(k)}\mathbf{I}_i}$ and $\boldsymbol{\phi}_{\mathbf{I}_i\mathbf{A}^{(k)}}$) between the current atlas $\mathbf{A}^{(k)}$ and the image $\mathbf{I}_i$. The latent representations of these transformations, $\mathbf{z}_{\mathbf{A}^{(k)}\mathbf{I}_i}$ and $\mathbf{z}_{\mathbf{I}_i\mathbf{A}^{(k)}}$, are extracted using the LEDA module. At each iteration, the latent representations corresponding to the transformations from the atlas to all images, $\mathbf{z}_{\mathbf{A}^{(k)}\mathbf{I}_i}$, are averaged across all $i$ to compute a mean representation, $\mathbf{z}^k$. This negated mean representation is decoded to obtain the atlas to image mean deformation field $\overline{\boldsymbol{\phi}}^k$, which is then used to update the atlas $\mathbf{A}^{(k)}$ pulling it towards the true mean. The process is repeated until the atlas converges to a stable solution, as determined by minimal changes across iterations. Additional details are provided in the Appendix. To evaluate the robustness of this method, we perform multiple estimations, each initialized with a different atlas. The initial atlases are chosen randomly with respect to two distinct age groups: above 45 years and below 45 years. As shown in Figure 5, the final estimated atlas remains an unbiased estimate regardless of the initialization, demonstrating the robustness of the approach. Unlike a naive pixel-wise average that may introduce artifacts, the proposed approach ensures consistent geometric alignment across the dataset. This alignment is crucial for studying population variability and enables effective downstream statistical analysis, ensuring that the atlas remains biologically meaningful and robust for comparative studies.

## 5. Conclusion

The proposed MORPH-LER framework serves as a plug-and-play solution, enhancing existing registration networks for improved anatomical accuracy and interpretability across synthetic and medical imaging datasets. It provides a compact statistical representation of deformation fields, aiding downstream tasks like morphological analysis and population studies. Future work could focus on incorporating diagnostic and clinical covariates to enhance the diffeomorphic analysis framework. MORPH-LER's inherent computational efficiency makes it a promising candidate for extension to 3D registration tasks.

## Acknowledgments

This work was supported by the National Institutes of Health under grant numbers NIH-R01CA259686 (Sarang Joshi) and NIH-R01DE032366 (Shireen Elhabian).

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

## 6. Appendix

### 6.1. Atlas Estimation Using the Fréchet Lie Group Structure

This algorithm utilizes the Lie group structure in the latent space of the trained MORPH-LERto achieve a geometrically consistent and computationally efficient framework for atlas estimation. The objective is to iteratively refine the atlas while ensuring its validity as a reference in the deformation space.

1. **Initialize the Atlas:** Randomly select an image from the input set $\{\mathbf{I}_1, \mathbf{I}_2, \ldots, \mathbf{I}_N\}$ as the initial atlas $\mathbf{A}^{(k)}$ for $k = 0$.

2. **Estimate Deformation Fields and Latent Representations:** For each input image $\mathbf{I}_i$, perform the following steps:

   (a) Compute the deformation field $\boldsymbol{\phi}_{\mathbf{I}_i\mathbf{A}}$ that aligns the input image $\mathbf{I}_i$ to the current atlas $\mathbf{A}^{(k)}$ using the registration module of the MORPH-LER framework. Similarly, compute $\boldsymbol{\phi}_{\mathbf{A}\mathbf{I}_i}$, which aligns $\mathbf{A}^{(k)}$ to $\mathbf{I}_i$.

   (b) Extract the latent representations $\mathbf{z}_{\mathbf{I}_i\mathbf{A}}$ and $\mathbf{z}_{\mathbf{A}\mathbf{I}_i}$ corresponding to the deformation fields $\boldsymbol{\phi}_{\mathbf{I}_i\mathbf{A}}$ and $\boldsymbol{\phi}_{\mathbf{A}\mathbf{I}_i}$, respectively, using the LEDA module of the trained MORPH-LER framework.

3. **Calculate the Mean Deformation Field:** Compute the mean of the latent representations $\{\mathbf{z}_{\mathbf{A}\mathbf{I}_1}, \mathbf{z}_{\mathbf{A}\mathbf{I}_2}, \ldots, \mathbf{z}_{\mathbf{A}\mathbf{I}_N}\}$ corresponding to the deformation fields:

$$\mathbf{z}^k = \frac{1}{N} \sum_{i=1}^{N} \mathbf{z}_{\mathbf{A}\mathbf{I}_i}.$$

   Use the LEDA module to map the mean latent representation $\mathbf{z}^k$ to the corresponding mean deformation field $\overline{\boldsymbol{\phi}}^k$ and negate $\mathbf{z}^k$ and decode it to get the inverse of the mean deformation field $\overline{\boldsymbol{\phi}}_{inv}^k$.

4. **Update the Atlas:** Warp the current atlas $\mathbf{A}^{(k)}$ using $\overline{\phi}_{inv}^{k}$ deformation field that brings the current atlas $\mathbf{A}^{k}$ closer to the mean of the population of images. The updated atlas is computed as:

$$\mathbf{A}^{(k+1)}(\mathbf{x}) = \mathbf{A}^{(k)}(\overline{\phi}_{\alpha}^{k}(\mathbf{x})). = \overline{\phi}_{inv}^{k}(\mathbf{A}^{k})$$

5. **Convergence Check:** Compute the change in the atlas between iterations:

$$\Delta = \frac{\|\mathbf{A}^{(k+1)} - \mathbf{A}^{(k)}\|}{\|\mathbf{A}^{(k)}\|}.$$

If the $\Delta$ is below a predefined threshold $\epsilon$, terminate the algorithm. Otherwise, increment the iteration index $k \leftarrow k + 1$ and repeat from Step 2.

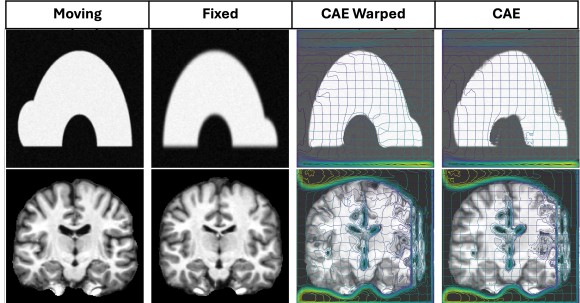

Figure 6: Figure shows the outputs of CoopNet (Bhalodia et al., 2019) for a toy dataset of a torus with a bump and the Oasis-2D dataset. While registering the moving image to the fixed image, the predicted deformation fields provide large over, and the transformations are not diffeomorphic.

