# OpenReview forum: "MORPH-LER: Log-Euclidean Regularization for Population-Aware Image Registration"
_MIDL.io/2025/Conference — MIDL 2025 Poster_

### Official Review · Reviewer_wWkS · 2025-02-18

**Confidence:** 4
**Preliminary Rating:** 5
**Recommendation:** Oral

**Summary:**

This paper introduces an algorithm for image registration and the creation of atlas from a population of 2d images, building on previous works. In particular, it combines a registration network (with two variants already published) and an additional autoencoder published in a separate paper by the authors (called LEDA) to constrain the first registration algorithm to preserve diffeomorphic properties of images so that they remain anatomically meaningfull.

**Strengths:**

The paper is generally well-written and self-contained in term of the authors LEDA's algorithm. The motivation is also clear and the results well-presented on a standard dataset, making it easy to understand.

**Weaknesses:**

One of the weakness in the presentation for me is the use of technical language, that makes is a little hard to understand for non-expert readers well aware of differential geometry concepts as well as the two registration methods. The paper may benefit from some small rewriting to make it more accessible, especially in the abstract/introduction, to use less technical words to present the problem and algorithm. Maybe also an short explanation on how the two registration algortithm work (maybe in method is space is too limited). It does not have to be long, just a paragraph to get the main difference in their inner working.

**Detailed Comments:**

I don't have any minor improvements besides the ones mentionned above.

**Justification Of The Preliminary Rating:**

I think it is a good paper on a fundamental topic in computational anatomy, that seems to produce relevant results on this stnadard dataset. It may have potential for future refinements and use on real data.

**Questions To Address In The Rebuttal:**

For me, imroving the text to make it more accessible would be sufficient, its a good paper otherwise!

**Special Issue:**

No

---

> ### Comment · Area_Chair_nmpU · 2025-02-23
> **Request for more technical details**
>
> Thanks for submitting your review. Could you please add some more technical details to support your rating?
>
> Right now, the review only talks about the paper's presentation, both as strengths (well-written/self-contained/clear/etc.) and as weaknesses (technical language being hard to understand). While presentation is important, it would be very helpful to include concrete technical details and evaluations of the soundness and depth of the paper's methodologies.
>
> The discussion period will benefit greatly from any time you can take to fill this out, thanks again for your time and effort!

---

> ### Author Response · Authors · 2025-03-07
> **Response to Reviewer wWkS**
>
> > Can you remove technical jargon so that non-experts familiar with differential geometry can understand? Also, can you briefly explain GradIcon and Transmorph in the method section, including their differences, in two sentences?
>
> Thank you for your comments regarding the use of technical language. We acknowledge that certain sections of the manuscript may benefit from simplification for a broader audience. We have revised the text to reduce the reliance on specialized terminology and provide clearer explanations of key concepts, while still maintaining the necessary rigor. Our goal is to make the work accessible to a wider range of readers without sacrificing technical accuracy. Also addressed the additional information about $\text{Transmorph}^1$, and $\text{GradICON}^2$.
>
> References:
>
> 1. Chen, Junyu, et al. "Transmorph: Transformer for unsupervised medical image registration." Medical image analysis 82 (2022): 102615.
>
> 2. Tian, Lin, et al. "GradICON: Approximate diffeomorphisms via gradient inverse consistency." Proceedings of the IEEE/CVF Conference on Computer Vision and Pattern Recognition. 2023.

---

### Official Review · Reviewer_JF4T · 2025-02-22

**Confidence:** 4
**Preliminary Rating:** 3
**Recommendation:** Poster

**Summary:**

The paper introduces a regularization technique that can be plugged into existing deep learning-based image registration methods. The primary purpose of the method is to regularize the deformation field to ensure diffeomorphic properties, while also facilitating further analysis of the deformation fields. Experiments on 2D sections of MRI brain images compare two deep learning-based registration methods, with and without the proposed regularizer, against a baseline method. The results suggest that the proposed technique improves the quality of the deformation field while maintaining competitive Dice scores.

**Strengths:**

Image registration is an ill-posed problem that requires regularization, and diffeomorphic properties are often used to validate and evaluate deformation fields. Having a method that can be incorporated into various approaches to improve warp field quality is both interesting and promising. The paper presents a well-motivated problem, and the proposed method could be beneficial for ensuring the stability and interpretability of learned deformation fields in deep learning-based registration.

**Weaknesses:**

While I am quite familiar with image registration, I may not have fully understood the deformation analysis presented in this paper. A more intuitive explanation would help clarify the method’s contributions. Additionally, the practical benefits over existing methods remain a bit unclear to me.

From my experience with methods such as LapIRN (Large Deformation Diffeomorphic Image Registration with Laplacian Pyramid Networks) and TransMorph (Transformer for Unsupervised Medical Image Registration), both already perform well in preserving diffeomorphic properties. LapIRN, in particular, did not produce negative Jacobians in my experiments. TransMorph penalizes them during training. Regularization techniques for the Jacobian determinant or smoothness constraints are commonly used, and I would have liked to see a comparison with such approaches. Additionally, classical methods like SyN from ANTs already enforce smooth and invertible deformations, which might provide another useful baseline.

Finally, I am concerned about computational costs and memory demands. The proposed approach requires computing both forward and backward transformations, in addition to an encoder-decoder network for the Diffeomorphic Autoencoder. Given that 3D image registration is already memory intensive, it is unclear whether this method is feasible for larger-scale applications.

**Detailed Comments:**

See: Questions To Address In The Rebuttal

**Justification Of The Preliminary Rating:**

I have rated this paper as Borderline due to concerns regarding the clarity of the method’s contributions, practical benefits over existing techniques, and computational feasibility for real-world applications. The proposed regularization technique is interesting, particularly because it can be integrated into various deep learning-based image registration methods. Ensuring diffeomorphic properties in deformation fields is an important problem, and the idea of also using the method for deformation analysis is promising.

However, I found aspects of the deformation analysis difficult to follow, and a more intuitive explanation of the equations would help clarify its significance. Additionally, while the method aims to improve deformation field quality, existing approaches such as LapIRN and TransMorph already achieve diffeomorphic properties or penalize negative Jacobians effectively. A comparison with a wider range of regularization methods, as well as traditional techniques like SyN (ANTs), would strengthen the paper.

Another concern is computational cost, especially for 3D medical images.

**Questions To Address In The Rebuttal:**

I understand that the method is intended for regularizing deformation fields during training, but it also seems applicable for analyzing deformation fields during inference. This aspect appears to be explored only superficially. Could the authors elaborate on its practical use for deformation field analysis?

The method appears computationally expensive. How feasible is it for 3D image registration? Many deep learning-based 3D registration methods already struggle with memory constraints, and this method requires additional transformations and an autoencoder. Have the authors evaluated its scalability?

 I would have liked to see a comparison with further regularization approaches.

---

> ### Author Response · Authors · 2025-03-07
> **Response to Reviewer JF4T - Part 1**
>
> > Can you provide a more intuitive explanation to clarify the method’s contributions? Also, could you highlight the practical benefits over existing methods more clearly?
>
> Thank you for your feedback. We recognize the need for a more intuitive explanation of MORPH-LER and a clearer articulation of its practical benefits over existing methods. MORPH-LER addresses a critical challenge in medical image registration: ensuring that deformations are both accurate and anatomically plausible. Existing methods often prioritize accuracy but may produce transformations that distort anatomical structures, leading to unreliable analyses. MORPH-LER overcomes this limitation through its unique approach:
>
> *Log-Euclidean Regularization with a Diffeomorphic Autoencoder $(LEDA^1)$*: This component learns the underlying patterns of anatomical variation within a population. By acting as a data-driven regularizer, it guides the registration process to produce transformations that are consistent with typical anatomical structures.
>
> *Population-Aware Registration*: By leveraging learned population information, MORPH-LER ensures anatomically consistent transformations, even for images with large differences. This reduces unrealistic deformations and improves the reliability of downstream analyses.
>
> *Linearized Latent Space*: MORPH-LER creates a compact and interpretable representation of deformation fields, improving efficiency while maintaining anatomical context.
>
> *Seamless Integration*: MORPH-LER can be easily incorporated into existing registration frameworks, allowing users to enhance their methods by simply plugging it in.
>
> **Practical Benefits**
>
> *More Reliable Image Analysis*: By reducing the risk of false positives in studies analyzing structural differences (e.g., patients vs. controls), MORPH-LER ensures that results are both accurate and anatomically meaningful.
> *Efficient Computation*: Its efficient encoding leads to fast and computationally lightweight analyses.
>
> By combining population-level information with a diffeomorphic framework, MORPH-LER offers a robust, accurate, and reliable approach to medical image registration, enabling more meaningful insights from medical imaging data. This improved understanding of MORPH-LER enhances the study of disease effects on image registration, leading to higher-quality, reproducible results and advancements in medical image analysis.
>
> References:
>
> 1. Iyer, Krithika, Shireen Elhabian, and Sarang Joshi. "LEDA: Log-Euclidean Diffeomorphic Autoencoder for Efficient Statistical Analysis of Diffeomorphism." arXiv preprint arXiv:2412.16129 (2024).
>
>
> > I understand that the method is intended for regularizing deformation fields during training, but it also seems applicable for analyzing deformation fields during inference. This aspect appears to be explored only superficially. Could the authors elaborate on its practical use for deformation field analysis?
>
> Thank you for your insightful comment. We believe MORPH-LER's key strengths lie in its ability to enhance both the accuracy of image registration and its potential for broader applications in population-level analysis and clinical settings, as we will elaborate below:
>
> MORPH-LER's ability to learn diffeomorphic transformations enhances the accuracy and consistency of image registration, which is particularly beneficial in population-level analysis. This enables precise quantification of anatomical variations across large datasets, providing crucial insights into normal anatomical variability and its relationship to factors like age or health conditions.
> Beyond improving image registration, MORPH-LER’s integration with clinical labels could enhance its ability to track subtle structural changes over time. This would be invaluable for monitoring neurodegenerative diseases such as Alzheimer’s or Parkinson’s, potentially leading to the identification of novel imaging-based biomarkers for early detection and prognosis prediction. Additionally, in clinical settings, the combination of anatomically consistent transformations with patient-specific clinical data could provide enhanced decision support for diagnosis and treatment planning.

---

> ### Author Response · Authors · 2025-03-07
> **Response to Reviewer JF4T - Part 2**
>
> > The method appears computationally expensive. How feasible is it for 3D image registration? Many deep learning-based 3D registration methods already struggle with memory constraints, and this method requires additional transformations and an autoencoder. Have the authors evaluated its scalability?
>
> Thank you for your insightful question regarding the extension of MORPH-LER to 3D. We appreciate your interest in this important aspect of the framework. We can confirm that extending MORPH-LER to 3D is indeed feasible and, importantly, retains its computational advantages. In fact, we have already successfully trained 3D versions of MORPH-LER, integrating $LEDA^1$ (Log-Euclidean Diffeomorphic Autoencoder), with both $GradICON^5$ on the $OASIS^2$ dataset and a toy dataset using a 40 GB GPU. The inherent design of $LEDA$, with its recursive network structure and weight reuse, allows for efficient management of computational resources, ensuring scalability even in 3D. While we recognize the value of demonstrating 3D results, we chose to focus primarily on the 2D implementation in this paper for strategic reasons:
>
> We encountered challenges in achieving optimal performance with the 3D $CAE^3$ network and $TransMorph^4$ which required significant fine-tuning and resource allocation.
>
> Given the time constraints, we prioritized a thorough and comprehensive analysis of the 2D implementation, allowing us to deeply explore the intricacies of MORPH-LER and its performance characteristics.
>
> We want to emphasize that we are confident in the potential of the 3D extension and view it as a promising avenue for future research. However, we believe that further optimization and analysis are necessary to present robust and convincing 3D results that meet our standards for scientific rigor. We plan to dedicate future work to fully exploring and validating the 3D capabilities of MORPH-LER.
>
> References:
>
> 1. Iyer, Krithika, Shireen Elhabian, and Sarang Joshi. "LEDA: Log-Euclidean Diffeomorphic Autoencoder for Efficient Statistical Analysis of Diffeomorphism." arXiv preprint arXiv:2412.16129 (2024).
>
> 2. Marcus, Daniel S., et al. "Open Access Series of Imaging Studies (OASIS): cross-sectional MRI data in young, middle aged, nondemented, and demented older adults." Journal of cognitive neuroscience 19.9 (2007): 1498-1507.
>
> 3. Bhalodia, Riddhish, et al. "A cooperative autoencoder for population-based regularization of CNN image registration." Medical Image Computing and Computer Assisted Intervention–MICCAI 2019: 22nd International Conference, Shenzhen, China, October 13–17, 2019, Proceedings, Part II 22. Springer International Publishing, 2019.
>
> 4. Chen, Junyu, et al. "Transmorph: Transformer for unsupervised medical image registration." Medical image analysis 82 (2022): 102615.
>
> 5.  Tian, Lin, et al. "GradICON: Approximate diffeomorphisms via gradient inverse consistency." Proceedings of the IEEE/CVF Conference on Computer Vision and Pattern Recognition. 2023.

---

> ### Author Response · Authors · 2025-03-07
> **Response to Reviewer JF4T - Part 3**
>
> > Could you include a comparison with additional regularization approaches, such as LAPIRN, which ensures diffeomorphic properties, and classical methods like SyN from ANTs, which enforce smooth and invertible deformations? Additionally, how does your method compare to common regularization techniques that enforce Jacobian determinant constraints or smoothness?
>
> Thank you for raising these important points regarding comparisons with other regularization approaches. We appreciate the suggestion to include $LAPIRN^1$ and $SyN^2$ (from ANTs) in our evaluation, as well as comparisons to methods directly regularizing the Jacobian determinant or smoothness.
>
> We acknowledge that methods like $SyN$, and more recently, $LAPIRN$, enforce smooth and invertible deformations. Similarly, techniques like Jacobian regularization or smoothness constraints are also common. $GradICON's^3$ extensive experiments demonstrated the benefits of gradient consistency over those traditional methods in registration. While important, existing regularizers fail to fully capture underlying population statistics, leading to anatomically inconsistent transformations. Inverse consistency regularizers promote geometric consistency, but they lack explicit integration of population morphometrics.
>
> However, MORPH-LER's core contribution is population-based regularization for image registration. We achieve this through $LEDA$, which offers several key advantages: (1) It learns a low-dimensional manifold that captures the essential modes of anatomical variation within a population, allowing us to effectively constrain and regularize the registration process. (2) It enforces diffeomorphic and latent inverse consistency, ensuring that the transformations are mathematically sound and biologically plausible. (3) As a byproduct of the $LEDA$ framework , we obtain an efficient way to estimate the principal logarithm of deformation fields, which is crucial for statistical analysis.
>
> A direct comparison with $LAPIRN$ and $SyN$  while valuable, would not fully showcase MORPH-LER's unique strengths in capturing and leveraging population-level information for diffeomorphic registration and not just enhance registration. We agree a comparison would be valuable but is beyond the scope of this work.
>
> That said, we recognize the value of situating MORPH-LER within the broader landscape of regularization techniques. In the revised manuscript, we will include a more detailed discussion of these methods in the related work section, highlighting their strengths and limitations in the context of population-aware registration. We will also clarify how MORPH-LER builds upon these approaches by introducing a novel framework that combines diffeomorphic properties with statistical shape analysis
>
> References:
>
> 1. Mok, Tony CW, and Albert CS Chung. "Large deformation diffeomorphic image registration with laplacian pyramid networks." Medical Image Computing and Computer Assisted Intervention–MICCAI 2020: 23rd International Conference, Lima, Peru, October 4–8, 2020, Proceedings, Part III 23. Springer International Publishing, 2020.
>
> 2.  Avants, Brian B., et al. "Symmetric diffeomorphic image registration with cross-correlation: evaluating automated labeling of elderly and neurodegenerative brain." Medical image analysis 12.1 (2008): 26-41.
>
> 3. Tian, Lin, et al. "GradICON: Approximate diffeomorphisms via gradient inverse consistency." Proceedings of the IEEE/CVF Conference on Computer Vision and Pattern Recognition. 2023.

---

> ### Comment · Area_Chair_nmpU · 2025-03-20
> **Final score**
>
> Dear reviewer: Could you please read the rebuttal and update your original review to include a final evaluation? Thanks!

---

> > ### Comment · Reviewer_JF4T · 2025-03-20
> > **Final score**
> >
> > Hi, I can't find an option to revise the text (final score) or a revised version of the paper. Any advice on how to get there would be appreciated.

---

> > > ### Comment · Area_Chair_nmpU · 2025-03-20
> > >
> > > I see, thank you for the response. I think the ability to edit reviews has been turned off now. If you leave an "official comment" on the paper with your final score and a sentence or two of justification, I can integrate it into the final metareview.

---

> > > > ### Comment · Reviewer_JF4T · 2025-03-21
> > > > **Here is my (late) final rating:**
> > > >
> > > > "4. weak accept"
> > > >
> > > > The paper raised many questions regarding the clarity of the method, computational costs, and limitations with respect to 3D images. I highly appreciate the authors’ detailed responses. I came to the conclusion that there are still several noteworthy limitations, particularly in 3D that should be mentioned in the paper. 3D is critical for many medical image-related tasks. However, I have not seen any attempts to include such information into the paper (Or I don't have access to a revised version). Overall, I would like to hear more about this methodology, so I will raise my rating to "4. weak accept," while not being fully confident with the paper content. Best wishes

---

### Official Review · Reviewer_4L1e · 2025-02-23

**Confidence:** 5
**Preliminary Rating:** 3
**Recommendation:** Poster
**Final Rating:** 4

**Summary:**

A new regularization framework is introduced, called MORPH-LER, which is a Log-Euclidean framework for population-aware unsupervised image registration. The two main methodological developments contributed by this submission are: data-driven regularization strategy, built on LEDA, and a linearized latent space representation. For the experiments, CAE, a cooperative autocencoder for population-based regularization, TransMorph and GradICON are used with 2D MRI slices from the OASIS dataset.

**Strengths:**

The submission is clearly written, it describes clear motivation and presents a clear intro section.

The framework presents a framework that allows for the computation of anatomically meaningful transformations with computational efficiency.

The framework builds on LEDA. This forms the secondary network along with the primary registration one.

**Weaknesses:**

The presentation of the results is a bit confusing and details are missing that would enable a more comprehensive understanding of how the experiments were run and evaluated.

The technical novelty is moderate due to the existing regularization framework and limited demonstration of the impact and superior performance of this framework.

The atlasing discussion in the Results section is a bit rushed and not fully developed.

The performance numbers are not as decisively convincing as discussed. Are the differences presented statistically significant? The four segmentation labels used provide a coarse metric for performance improvement.

**Detailed Comments:**

The figures are too dense and hard to parse. In Fig 1 the subparts are hardly visible, in Fig 2 they are not present. Jumping back to Fig 1 for the final conclusion at the end is a bit disorienting.

The framework is demonstrated in 2D. Is the 3D extension trivial? Does it still possess the same computational advantages?

What is the definition of an atlas that the authors consider? What is the desired atlas property? And what does consistency mean here? Pixelwise averaging is a very basic baseline to set given all other advanced atlas creation tools. Are the initial atlases single individuals? How were they chosen? Randomly?

Figs 2 and 3 are very busy and not well labelled. In the former the (a)-(c) labels are hard to find and inconsistently positioned in the latter the labels are not present. In Fig 3 the red arrows are not explained.
Fig 3 : What are the peppery voxels in the middle of the morphed slices? They seem like interpolation artefacts, but not likely in the middle of segmentation labels. Also lots of colors are used for col headers but they also coincide with segmentation labels. This makes the figure even more confusing.
Figure 4 is referenced in the main paper as a regular figure, but as the figure itself is not part of the 8-page limit, could this be relabeled as Appendix / Suppl?

Figure 5 is not referenced in the text

Code sharing statement is not in the text.

**Justification Of The Final Rating:**

The authors did a good job answering most of my questions and those of the other reviewers. I still have some concerns about the actual future impact of the proposed framework and its 3D implementation, but due to all the clarifications I will bump my final score to weak accept (poster).

**Justification Of The Preliminary Rating:**

The results are preliminary and lack in details so full evaluation is challenging. The performance improvement is not yet convincing, based on the currently presented results, that it would make significant improvement in  population atlasing as is proposed.

**Questions To Address In The Rebuttal:**

The segmentation labels seem combined compared to standard FS or AAL labels. Is it a 4-label segmentation that is used? Can the authors include details on these so that the evaluation of the Dice overlap metrics can be better interpreted?

Could you indicate where the differences between the various metrics are statistically different?

Could the figures be improved in  way that the key information is better highlighted and color-coding / arrows are explained?

Could the authors comment on whether the 3D extension of the proposed framework trivial? Does it still possess the same computational advantages?

Could the authors comment on what the desired atlas property are that they are optimizing? And what does consistency mean in the given setting?

---

> ### Author Response · Authors · 2025-03-07
> **Response to Reviewer 4L1e - Part 1**
>
> > How much Impact this can have on real world by learning diffeomorphic properties?
>
> Thank you for your insightful comment. We believe MORPH-LER's key strengths lie in its ability to enhance both the accuracy of image registration and its potential for broader applications in population-level analysis and clinical settings, as we will elaborate below
> MORPH-LER's ability to learn diffeomorphic transformations enhances the accuracy and consistency of image registration, which is particularly beneficial in population-level analysis. This enables precise quantification of anatomical variations across large datasets, providing crucial insights into normal anatomical variability and its relationship to factors like age or health conditions.
> Beyond improving image registration, MORPH-LER’s integration with clinical labels could enhance its ability to track subtle structural changes over time. This would be invaluable for monitoring neurodegenerative diseases such as Alzheimer’s or Parkinson’s, potentially leading to the identification of novel imaging-based biomarkers for early detection and prognosis prediction. Additionally, in clinical settings, the combination of anatomically consistent transformations with patient-specific clinical data could provide enhanced decision support for diagnosis and treatment planning.
>
>
> > Statistical significance of the results
>
> Thank you for your question regarding the statistical significance of the evaluation metrics for segmentation. To ensure a rigorous comparison, we assessed the statistical significance of our results by conducting pairwise statistical comparisons between our proposed methods (M-TransMorph and M-GradICON) and the baseline models ($CAE^1$, $Transmorph^2$, and $GradICON^3$). Specifically, we performed hypothesis testing using a paired statistical test, computing p-values for the metrics shown in Table 1, using paired t-tests. The results indicate that, for all model comparisons, the p-values are consistently below 0.05 across all held-out test samples. This confirms the differences in metrics (Dice Scores, % of Negative Jacobian determinant) reported in Table 1 are statistically significant, validating the improvements achieved by our proposed methods.
>
> References:
>
> 1. Bhalodia, Riddhish, et al. "A cooperative autoencoder for population-based regularization of CNN image registration." Medical Image Computing and Computer Assisted Intervention–MICCAI 2019: 22nd International Conference, Shenzhen, China, October 13–17, 2019, Proceedings, Part II 22. Springer International Publishing, 2019.
>
> 2. Chen, Junyu, et al. "Transmorph: Transformer for unsupervised medical image registration." Medical image analysis 82 (2022): 102615.
>
> 3. Tian, Lin, et al. "GradICON: Approximate diffeomorphisms via gradient inverse consistency." Proceedings of the IEEE/CVF Conference on Computer Vision and Pattern Recognition. 2023.
>
> > In Fig 3 the red arrows are not explained. Fig 3 : What are the peppery voxels in the middle of the morphed slices? They seem like interpolation artifacts, but not likely in the middle of segmentation labels.
>
> Thank you for your questions regarding Figure 3. We appreciate the opportunity to clarify the details of the visualizations. In Figure 3.A, which illustrates the MORPH-LER PCA modes of logarithm maps and Latent Representations, the red arrows highlight key structural shape changes associated with each PCA mode. These arrows are intended to guide the viewer's attention to the primary anatomical variations captured by that specific mode. Regarding the appearance of the morphed slices, we understand why the voxels might appear as interpolation artifacts at first glance. However, they represent a visualization of the deformation field. To illustrate the modes of variation in the deformation field PCA, we display the deformation field itself and apply it to a randomly chosen image. This visualization provides insight into the local transformations, representing the complex spatial relationships captured by the PCA modes. We acknowledge that this visualization approach may require careful interpretation and will consider alternative representations in future iterations of this work to improve clarity.
>
> > Code sharing statement is not in the text.
>
> We appreciate the reviewer's observation. We have added the GitHub repository in the abstract as a footnote to ensure reproducibility and transparency of our research.

---

> ### Author Response · Authors · 2025-03-07
> **Response to Reviewer 4L1e - Part 2**
>
> > Could the authors comment on whether the 3D extension of the proposed framework trivial? Does it still possess the same computational advantages?
>
> Thank you for your insightful question regarding the extension of MORPH-LER to 3D. We appreciate your interest in this important aspect of the framework. We can confirm that extending MORPH-LER to 3D is indeed feasible and, importantly, retains its computational advantages. In fact, we have already successfully trained 3D versions of MORPH-LER, integrating $LEDA^1$ (Log-Euclidean Diffeomorphic Autoencoder), with both $GradICON^5$ on the $OASIS^2$ dataset and a toy dataset using a 40 GB GPU. The inherent design of $LEDA$, with its recursive network structure and weight reuse, allows for efficient management of computational resources, ensuring scalability even in 3D. While we recognize the value of demonstrating 3D results, we chose to focus primarily on the 2D implementation in this paper for strategic reasons:
>
> We encountered challenges in achieving optimal performance with the 3D $CAE^3$ network and $TransMorph^4$ which required significant fine-tuning and resource allocation.
>
> Given the time constraints, we prioritized a thorough and comprehensive analysis of the 2D implementation, allowing us to deeply explore the intricacies of MORPH-LER and its performance characteristics.
>
> We want to emphasize that we are confident in the potential of the 3D extension and view it as a promising avenue for future research. However, we believe that further optimization and analysis are necessary to present robust and convincing 3D results that meet our standards for scientific rigor. We plan to dedicate future work to fully exploring and validating the 3D capabilities of MORPH-LER.
>
> References:
>
> 1. Iyer, Krithika, Shireen Elhabian, and Sarang Joshi. "LEDA: Log-Euclidean Diffeomorphic Autoencoder for Efficient Statistical Analysis of Diffeomorphism." arXiv preprint arXiv:2412.16129 (2024).
>
> 2. Marcus, Daniel S., et al. "Open Access Series of Imaging Studies (OASIS): cross-sectional MRI data in young, middle aged, nondemented, and demented older adults." Journal of cognitive neuroscience 19.9 (2007): 1498-1507.
>
> 3. Bhalodia, Riddhish, et al. "A cooperative autoencoder for population-based regularization of CNN image registration." Medical Image Computing and Computer Assisted Intervention–MICCAI 2019: 22nd International Conference, Shenzhen, China, October 13–17, 2019, Proceedings, Part II 22. Springer International Publishing, 2019.
>
> 4. Chen, Junyu, et al. "Transmorph: Transformer for unsupervised medical image registration." Medical image analysis 82 (2022): 102615.
>
> 5.  Tian, Lin, et al. "GradICON: Approximate diffeomorphisms via gradient inverse consistency." Proceedings of the IEEE/CVF Conference on Computer Vision and Pattern Recognition. 2023.
>
>
> > The segmentation labels seem combined compared to standard FS or AAL labels. Is it a 4-label segmentation that is used? Can the authors include details on these so that the evaluation of the Dice overlap metrics can be better interpreted?
>
> Thank you for your question regarding the segmentation labels used in our evaluation. We understand the importance of clarifying these details for proper interpretation of the Dice overlap metrics.
>
> Figure 3. B showcases a simplified 4-label brain segmentation used to assess registration quality. The segmentation comprises cortex, gray matter, sub-cortical gray matter, and cerebrospinal fluid, providing a concise yet informative representation of key brain structures. To evaluate registration performance, we apply learned deformation fields to these segmentation labels and compute Dice scores between the warped and target segmentations. This quantitative approach enables us to measure how accurately the registration aligns critical brain structures across images. The accompanying table in the figure presents Dice scores for each of the four classes, specifically for the visualized sample. For clarity, the colors used in the segmentation image correspond directly to those in the table, albeit with higher opacity. In Table 1, we report the registration performance across all test samples. The second column presents the overall average Dice score across all four classes, while the subsequent columns provide class-specific Dice scores for each model. These scores are computed using all the test samples, which were held out during training, providing a robust evaluation of each model's generalization capability and registration accuracy across different brain structures.
>
> We have changed the text to improve the explanation of these details in the manuscript.

---

> ### Author Response · Authors · 2025-03-07
> **Response to Reviewer 4L1e - Part 3**
>
> > What is the definition of an atlas that the authors consider? What is the desired atlas property? And what does consistency mean here? Pixelwise averaging is a very basic baseline to set given all other advanced atlas creation tools. Are the initial atlases single individuals? How were they chosen? Randomly?  Could the authors comment on what the desired atlas property is that they are optimizing? And what does consistency mean in the given setting?
>
> We are following the well-established definition of an Atlas within Computational $Anatomy^1$: More formally the atlas is a statistical model representing the average shape, and structure, of a population of anatomical objects. It's a reference or coordinate system to study and compare anatomical variability across individuals. To accomplish this Atlas needs to be unbiased and independent of the initial choice.
>
> Thank you for the question. We validated this by initializing the atlas with single scans from different individuals randomly selected from distinct age groups (one above 45 and one below 45). We intentionally selected individuals from these age groups to represent a broad spectrum of anatomical variability, particularly given the known differences in ventricular size and brain structure between younger and older adults, providing significantly different starting atlases. Multiple experiments with varying initializations demonstrated that the final estimated atlas was not biased towards initial choice, and that the method reliably converges to a stable solution.
>
> The optimization process ensures that the atlas remains a valid reference in the deformation space, preventing the artifacts and smoothing effects that can arise from simple averaging techniques. By leveraging the Lie group structure in the latent space, our approach guarantees a meaningful and geometrically consistent alignment across the dataset. This results in a reliable and biologically meaningful atlas suitable for comparative population studies.
> We have included more details on the atlas estimation process in the revised manuscript. We also updated the explanation of the consistency to clarify its meaning, as it could be easily misunderstood due to its usage in other contexts. We have updated the existing explanation to make it clearer.
> References:
> 1. Joshi, Sarang, et al. "Unbiased diffeomorphic atlas construction for computational anatomy." NeuroImage 23 (2004): S151-S160.

---

> ### Comment · Reviewer_4L1e · 2025-03-12
>
> Thanks for the detailed response of the authors to both my and the other reviewers' comments.
>
> * re impact: the future impact, I agree could be significant, but it seems to still be more hypothetical at this point of development.
>
> * re stat significance: I appreciate the added computations and clarifications
>
> * re morph visualization: this still remains a bit unresolved and inconsistent as the label maps are used for downstream Dice computations; this section / figure could still create confusion with readers
>
> * re code: github link noted
>
> * re 3D: noted; I would have probably included a bit more of the answer in the submission (compared to the half sentence) though
>
> * re seg labels / Dice overlap: I appreciate the added detail
>
> * re atlas / biasedness: thanks for adding definition and experimental details; What are the ways the authors compute biasedness? I see qualitative differences between the atlases displayed in Fig 5, but biasedness (or lack of it) is hard to assess. Both sets of resulting atlases have more resemblance to their initial images. Is the biasedness evaluated within the group of younger and older subjects or across these groups?

---

> > ### Author Response · Authors · 2025-03-13
> > **Response to Reviewer 4L1e - Part 4**
> >
> > > Re Impact: the future impact, I agree could be significant, but it seems to still be more hypothetical at this point of development.
> >
> > Thank you for your feedback—we truly appreciate your perspective on the potential impact of MORPH-LER. We agree that broader clinical integration is still an ongoing process. However, our current results already demonstrate practical benefits, including atlas estimation and latent traversals, which serve as a solid foundation for population studies beyond just theoretical implications.
> >
> > While the long-term clinical impact—such as biomarker discovery and disease monitoring—will require further validation through longitudinal studies and clinical collaborations, our findings strongly indicate its feasibility. We are actively working on validating these aspects using additional real-world datasets.
> >
> > We'd be happy to discuss this further and appreciate the opportunity to do so.
> >
> > > Re 3D: noted; I would have probably included a bit more of the answer in the submission (compared to the half sentence) though.
> >
> > We appreciate your feedback on the 3D aspect. In the camera-ready version, we will expand on this discussion in the paper to provide a more complete explanation as discussed above.
> >
> >
> > > Re atlas / biasedness: thanks for adding definition and experimental details; What are the ways the authors compute biasedness? I see qualitative differences between the atlases displayed in Fig 5, but biasedness (or lack of it) is hard to assess. Both sets of resulting atlases have more resemblance to their initial images. Is the biasedness evaluated within the group of younger and older subjects or across these groups?
> >
> > We appreciate your question. The evaluation of bias was conducted within each age group (younger and older subjects) based on different initializations from individuals belonging to those groups—not across age groups. To assess bias more rigorously, we are refining our approach and designing additional experiments as part of our future work.
> >
> > We acknowledge that assessing bias from Figure 5 alone can be challenging. To evaluate this, we conducted multiple atlas initializations within each age group and consistently observed convergence to similar atlases. Our primary goal with this figure is to illustrate that our approach ensures meaningful and geometrically consistent alignment within each age group, resulting in a biologically meaningful atlas. Despite starting from different initial atlases (ages 23 and 89), both groups converge to representative mean atlases that serve as reliable references for their respective populations. These atlases provide a strong foundation for statistical analysis within each group.
> >
> > Additionally, the differences observed between the estimated MORPH-LER atlases for ages 23 and 89 reflect structural variations between the groups. This highlights that the learned latent space effectively captures distinct anatomical characteristics, leading to separate yet meaningful atlas representations for each age group.

---

### Author Rebuttal · Authors · 2025-03-07

**Rebuttal:**

We sincerely thank the reviewers for their insightful comments and valuable feedback on our paper. We have carefully addressed the concerns raised to enhance the clarity, rigor, and comprehensibility of our work.

The reviewer's concerns are addressed in the attached revised paper while ensuring the content remains concise and well-structured. If any questions are not answered, we will incorporate them into the camera-ready submission.

Addressed comments in the manuscript are highlighted in red.

**Supporting Material:**

/attachment/876a4682687c786cdd908933d295e6aeb41045f7.pdf

---

### Comment · Area_Chair_nmpU · 2025-03-12
**Discussion period**

This paper initially received borderline scores from its two most detailed reviews, with important issues being raised. These included no comparisons with popular work (e.g., SyN, LapIRN, etc.), marginal gains, and unclear application to real 3D volumes (instead of toy slices). With the posted rebuttal, would anyone like to change their mind and/or advocate for this paper?

Either way, please take a few moments to consider the paper's rebuttal arguments, clarify any remaining questions in the discussion period, and enter in your final evaluation. It would be great to build towards a consensus. Thanks again for your time and expertise!

---

### Meta-Review · Area_Chair_nmpU · 2025-03-21

**Recommendation:** Accept (Poster)
**Confidence:** 4

**Metareview:**

This paper received three reviews. To summarize its weaknesses:
- Reviewer 4L1e initially rated it as a borderline submission raising important concerns such as unclear results presentation, a lack of technical novelty, and no clear performance improvements. Post rebuttal, they raised their score to a weak accept with reservations pertaining to future applicability to real 3D data.
- Reviewer JF4T also initially considered it to be a borderline submission, with weaknesses including a lack of baseline comparisons to other diffeomorphic methods, higher computational demands, and confusing presentation. Post rebuttal, due to missing the openreview window, they were unable to update their original rating, but raised to a weak accept in correspondence to the AC and PC with their remaining concern being a lack of discussion around application to 3D data.
- Lastly, reviewer wWkS did strongly accept the paper, but unfortunately, this review was lacking in technical detail and self-contradictory ("strengths: well presented; weaknesses: hard to understand") and they did not interact beyond the initial review.

However, reviewers 4L1e and JF4T both appreciated the problem setup and formulation, thought the motivation was interesting, and found the rebuttal responses to be convincing. On net balance, this paper should be accepted.